# Genital Wound Repair and Scarring

**DOI:** 10.3390/medsci10020023

**Published:** 2022-04-18

**Authors:** Ursula Mirastschijski, Dongsheng Jiang, Yuval Rinkevich

**Affiliations:** 1Mira-Beau Gender Esthetics Berlin, 10777 Berlin, Germany; 2Wound Repair Unit, CBIB, Department of Biology and Biochemistry, University of Bremen, 28359 Bremen, Germany; 3Comprehensive Pneumology Center, Institute of Lung Biology and Disease, Helmholtz Zentrum München, 81377 München, Germany; dongsheng.jiang@helmholtz-muenchen.de (D.J.); yuval.rinkevich@helmholtz-muenchen.de (Y.R.); 4Institute of Regenerative Biology and Medicine, Helmholtz Zentrum München, 81377 München, Germany

**Keywords:** genital skin, foreskin, small and big labia, genital wound healing, genital scarring, lichen sclerosus, genital biomechanics, intracrine hormone production

## Abstract

Skin wound repair has been the central focus of clinicians and scientists for almost a century. Insights into acute and chronic wound healing as well as scarring have influenced and ameliorated wound treatment. Our knowledge of normal skin notwithstanding, little is known of acute and chronic wound repair of genital skin. In contrast to extra-genital skin, hypertrophic scarring is uncommon in genital tissue. Chronic wound healing disorders of the genitals are mostly confined to mucosal tissue diseases. This article will provide insights into the differences between extra-genital and genital skin with regard to anatomy, physiology and aberrant wound repair. In light of fundamental differences between genital and normal skin, it is recommended that reconstructive and esthetic surgery should exclusively be performed by specialists with profound expertise in genital wound repair.

## 1. Introduction

In contrast to injuries of the limbs, face or trunk, genital tissue defects due to trauma are rare. Results of literature searches for wound repair of genital tissue refer mostly to the urinary tract or vagina but not the skin. This is intriguing since male circumcisions have been practiced by many communities since ancient times. Little is known of the incidence and severity of scarring in the genital area despite the fact that 30% of all men are circumcised [1] (Figure 1). Current body culture involving shaving of the intimate areas has an impact on the desired esthetic changes of the outer genitalia in men and women [2]. Labioplasty, for example, is performed with increasing frequency, with almost no visible scarring. In recent decades, gender reassignment surgery has attracted great interest, with focus on functional as well as esthetic reconstruction. Despite massive anatomical changes, genital scarring is minimal in both reassigned genders (Figure 1).

Burn or combat related genital injuries can cause wounds and scarring to genital organs. Incidence of burn injury to the genitalia is rare at around 1.5% [3]. Burn patients who also had a burn injury to the genitalia showed a higher mortality rate of 17% in comparison to patients without genital burns (4.7%) [4]. Hypertrophic scarring is rarely seen in outer genital organs.

Infections are quite common in the genital area. In case of diabetes or immune incompetence of the host, germs have access to deeper tissue layers through small wounds and thereby cause severe infections, e.g., abscesses or gangrene with high mortality rates. Patients with hidradenitis suppurativa have chronic infections of the sweat glands and hair follicles. In this chronic disease, patients develop fistulas with recurrent infections. The annual prevalence of hidradenitis suppurativa is around 1% with 70 million diseased patients globally [5].

Fournier gangrene is another infectious disease of the genital skin. This life-threatening disease occurs with an incidence of 1.6/100,000 patients and is mostly present in men (10:1) [6]. The only sufficient cure is thorough surgical debridement of necrotic tissue with simultaneous antibiotic treatment. This is followed by massive tissue loss and subsequent plastic-surgical reconstruction [7]. In general, plastic surgical treatment follows an algorithm for defect coverage of various sizes. For perineal wounds, such an algorithm was published by Sharma et al. 2012 [8].

Chronic genital wounds are present in various dermatological diseases, e.g., lichen sclerosus et atrophicus (LSC). LSC is an autoimmune disease with chronic inflammation of the mucous surfaces of the anogenital area [9] which affects both sexes and can lead to severe scarring and tissue shrinkage when untreated [9].

The genital area has urogenital and anal orifices which are colonized by resident microbia in a moist environment. Skin tissue is found in abundance with highly elastic skin that is loosely attached to the pelvic bone. Hence, genital skin differs in several aspects from non-genital skin. The inflammatory reaction and hormonal responsiveness of genital skin differ in comparison to other tissues due to cellular responses adapted to the local microflora, with fast resolution of inflammatory responses and wound closure.

The aim of this review article is to summarize the current knowledge on wound repair of genital skin and compare to non-genital skin. Furthermore, we add histological analyses of genital and non-genital skin from our own patient cohort to provide evidence on anatomical differences between the skin of different body sites.

## 2. Differences between Genital and Non-Genital Skin

### 2.1. Developmental Genital Similarities between Sexes

Although the macroscopic view of female and male genitalia differs tremendously, the histology of corresponding anatomical entities is very similar. In order to understand the similarities, one has to consider the developmental origin of the corresponding genital feature. The external sex organs develop from genital buds. Under hormonal influences, these buds differentiate into penis or small labia, glans or clitoris, scrotum or big labia (Table 1). Aside from developmental similarities between sexes, skin tissue provenience is important. The mucosa of the inner part of the vulva originates from the embryonic endoderm with higher hydration, lower friction permeability and lower irritant susceptibility compared to forearm skin [10].

### 2.2. Cutaneous Microstructure, Biomechanics and Biochemical Composition

The cutaneous extracellular matrix is predominantly composed of collagen type I (90%), and to a lesser degree of collagen type III, which is increased during wound repair and regeneration. Collagen, elastin and fibronectin are the main fibers constituting the connective tissue of the skin, providing an efficient barrier and safety against tension and shear forces. The skin is structured into the epidermis at the air-liquid interphase, the dermis with the papillary, reticular dermal layers and superficial fascia layer underneath containing vessels, nerves, sweat and sebaceous glands as well as the hair follicles [12]. In non-genital skin, the subcutaneous fat is adjacent to the reticular dermal layer. Interestingly, the subcutaneous fat itself is separated into an upper and a lower compartment by the Scarpa fascia (fascia cutanea superficialis). The Scarpa fascia is an evolutionary remnant of the carnosus muscle in animals. In humans, it is still present as the platysma in the neck and in the genitalia. In men, it comprises the Dartos muscle in the scrotum or the Dartos fascia, localized superficially underneath the skin in the penile shaft (Figure 2). In women, it is named Colles’ fascia and is found in the labia. This so-called Dartos “fascia” underneath the penile skin has carnosus muscle origin, and in fact is a “mistake” in the conventional nomenclature. These tissues are completely different from the superficial fascia beneath the non-genital skin, which mainly contains fascia fibroblasts instead of muscle cells, and has been shown to be pro-scarring [12]. Blocking fascia mobilization in non-genital skin has been shown to effectively reduce scarring after wound repair [13,14]. The presence of a highly elastic striated muscle and the lack of an anchoring structure, e.g., fat, in the outer genitals is the underlying reason for differences with regard to genital biomechanics in wound repair and scarring in comparison to non-genital skin. As seen in animals, genital tissue heals primarily by wound contraction, shrinkage and with almost no scarring [15].

The biochemical composition of the skin and its different layers is well established [16,17]. The dermal extracellular matrix (ECM) consists predominantly of different types of collagens, followed by elastic fibers, glycoproteins and proteoglycans. Amongst collagens, type I collagen is abundantly produced by dermal fibroblasts in adult skin. Fibroblasts are the predominant cell type in the connective tissue that produce various extracellular matrix proteins, remodel the ECM constantly and are important for wound repair. As tissues of different origins are exposed to tissue-specific biomechanical cues, fibroblasts originating from various body sites behave differently. Dermal fibroblasts were very sensitive to stiff matrices whereas foreskin fibroblasts did not increase the production of collagen type III, matrix metalloproteinase (MMP)-2 or tissue inhibitor of MMP-2 [18] in response to mechanical strain. Importantly, cellular MMP secretion is crucial for tissue remodeling and cell migration in skin wound repair [19,20,21].

Elastic fibers comprise elastin and microfibrils. While collagen fibers contribute to the tensile strength of the dermis, elastin is important for tissue elasticity and protects against tearing stresses [17]. Interestingly, the percentage of elastic fibers varies between skin tissue from various sites. While dermal elastin content is around 2% to 8% [22], genital skin contains up to 29% elastin [23]. Both elastic fibers and collagens are degraded by various MMPs [24,25]. The fragmentation of elastic fibers is found in aged skin [26] and fibrotic diseases such as lichen sclerosus [27]. 

In agreement with the literature [23,28], we found abundant elastic fibers in genital tissue in contrast to non-genital skin in our patient cohort which underwent plastic reconstructive surgery after urological interventions (biological men and women), gender reassignment surgery (trans-males and trans-females) and aesthetic genital surgery, with a total of 68 patients [11] (Figure 3).

### 2.3. Macroscopic and Microscopic Anatomy of Genital Skin

Penile and labial (small labia and inner aspect of the big labia) skin is devoid of hair follicles and sebaceous and sweat glands in contrast to normal skin. The glans penis and clitoris, the inner part of the foreskin (both sexes) and the small labia are covered by non-keratinizing epithelium, and thus resemble the oral cavity. There are many additional parallels between genital and oral wound repair which will be discussed later. The genital dermis is rich in loosely packed elastic fibers (Figure 3) to enable cutaneous adaptation to volume changes during erection, genital intercourse or childbirth. Of note, the penile size increases during erection within seconds by about 1.4 to 1.6-fold in length (unpublished data, [29]) and about 1.3-fold in circumference on average [29,30]. Excess skin with high elasticity and several fasciae allows for fast volume changes and tissue extension.

### 2.4. Genital Autocrine Hormone Production and Endocrine Responsivness

Skin wound repair is influenced by hormones. While estrogens accelerate wound healing [31], testosterone has the opposite effect [32]. In postmenopausal women and in men, adipose tissue and skin are the major sources of peripheral estrogen production. Skin cells can produce their own sex hormones by converting renal precursor peptides such as dehydroepiandrosterone (DHEA) into estrone by aromatase or into testosterone by 17β-hydroxysteroid-dehydrogenase (17β-HSD) (Figure 4). Estrogens enhance epidermal keratinocyte and dermal fibroblast migration [33]. In non-genital skin, steroid hormone addition stimulated fibroblast contraction without alpha-smooth-muscle expression [33], whereas the contractility of tunica albuginea fibroblasts was directly enhanced by estradiol with concomitant reduction of myofibroblasts [34]. Proliferation was increased in non-genital [35] and genital [36] keratinocytes by estradiol via estrogen receptor (ER) signaling. ERα and ERβ receptors are expressed both in non-genital and genital skin [36,37,38]. The anti-inflammatory effect of estrogens has been extensively demonstrated for non-genital skin [32,39]. In foreskin keratinocytes, anti-inflammatory estradiol reduced TNFα induced RANTES secretion, the effects of monocyte chemoattractant protein-1, and inhibited the effect of IFNγ [40,41,42].

Intracrine production of estrogens requires the enzyme aromatase which converts C19 steroid hormone precursors, e.g., androstenedione or testosterone, into estrone. In contrast to dermal fibroblasts, keratinocytes increase their aromatase secretion 400-fold upon mechanical wounding with subsequently enhanced estrogen levels [33]. Estrogens reduce the inflammatory reaction [43], oxidative stress and cellular apoptosis during wound repair [11,44]. Aromatase activity is age-dependent [45] and expressed in genital and non-genital fibroblasts [46,47]. In postmenopausal women, aromatase is upregulated in skin fibroblasts [48] and catalyzes estrogen production via testosterone aromatization [47]. Alternatively, testosterone is irreversibly converted into dihydrotestosterone (DHT) by 5α-reductase [49]. Interestingly, testosterone metabolism occurs much faster in genital skin fibroblasts compared to non-genital cells [50]. 5α-reductase is expressed in genital and non-genital skin [51] with age-dependent decrease of the type 2 but not type 1 enzyme in human foreskin [52]. DHT acts via binding to the intracellular androgen receptor (AR) which is found in both genital and non-genital skin [47]. DHT exposure increased binding to AR in genital skin fibroblasts in a dose- and time-dependent manner [53]. Significantly higher levels of AR, 5α-reductase and 17β-HSD were found in scrotal fibroblasts compared to blood cells [51]. Further information on hormone production and sex hormone receptor expression between non-genital and genital skin are found in detail elsewhere [11,15].

### 2.5. Normal Genital Wound Healing

Genital wound healing proceeds in general uneventfully and quickly—even though the skin and the mucous membranes of the genitalia are inhabited by many different commensal microbia. If one considers tissue trauma during childbirth, fast and uncomplicated wound repair is of vital importance for the survival of the species. Typically, an initial massive swelling is observed after genital trauma with fast resolution and almost invisible scarring after wound closure is accomplished. Hypertrophic scarring—as found in non-genital skin—is extremely rare in genitalia.

Excess of highly elastic and flexible skin is an important feature of the outer genitalia in both sexes. This phenomenon is very important not only for fast volume changes during sexual intercourse, thermal regulation of the testes or childbirth, but also has implications for reconstructive purposes. In case of traumatic tissue loss, half of the scortal sac can be reconstructed from the remaining scrotal tissue [11], which is highly similar to the wound healing characteristics of the spiny mouse Acomys [54]. Furthermore, excess skin is useful for defect coverage of the penile shaft in obese men [55,56], after trauma [57] or tumor excision [30,57] or for genital reassignment surgery [58]. Of note, skin adjacent to the outer genitalia, e.g., groin, suprapubic or perineal skin in the vicinity of the thigh, have a tendency to excessive scarring, whereas hypertrophic scarring is rarely found in genital skin [15]. In contrast, tissue shrinkage is typical after genital trauma. To summarize, diverging biomechanics and the cellular immune response of genital skin in comparison to non-genital skin support fast and almost scar-less wound healing of the genitalia.

### 2.6. Pathological Wound Healing of the Genital Skin

As pointed out above, acute genital wound repair is amazingly fast and uneventful in contrast to chronic repair processes in that area. Delayed wound healing after trauma, burns, massive tissue infections such as Fournier gangrene [7], foreign body granuloma or autoimmune diseases can cause chronic repair processes in the genitalia. Chronic genital inflammation can be caused by foreign substances, such as silicone, paraffin, Vaseline and many others, injected underneath the penile skin for enlargement purposes. Granuloma formation is followed by chronic inflammatory response and subsequent ulcer formation and tissue loss. The only rescue is the radical excision of all foreign body material and plastic surgical tissue reconstruction [30,57].

Autoimmune diseases of the genitalia include, amongst others, Lichen sclerosus et atrophicus (LSC, also named Balanitis xerotica obliterans in men), Lichen planus (almost indistinguishable from LSC [59]), Behçet’s and Crohn’s disease. Of all these diseases, LSC is most commonly found in genital skin with a ratio of 10:1 for women compared to men. In both sexes, chronic inflammation leads to skin atrophy and phimosis in men or tissue shrinkage in women followed by itchiness, pain and dyspareunia. An increased autoimmune response with formation of auto-antibodies against ECM-1 protein are features of this multifactorial disease [60]. Interestingly, a higher dermal inflammatory infiltrate with immune cells degrading elastic fibers was found in extra-genital LSC compared with genital skin [61]. LSC is associated with the development of squamous cell carcinoma in women in about 5% [62] and in men up to 30% [63]. In Behçet’s disease, the permanent presence of M1 macrophages sustains a chronic vasculitis with ulcer formation in genital and oral mucous epithelia [64].

## 3. Discussion

A great body of literature is available on skin wound healing and scarring derived from basic research and clinical studies. When scrutinizing the available data, very little information is available on genital wound repair. This is even more surprising if one considers that gynecologists, urologists, general and plastic surgeons regularly perform surgery in the genital and adjacent areas. In recent decades, gender reassignment and reconstructive surgery [58] after female genital mutilation/cutting [65] have gained more importance in plastic surgery. The aesthetic focus has shifted towards the genitalia with many women and men shaving their intimate areas [2]. Due to the awareness of the visible outer genitalia, now devoid of hair, the number of genital aesthetic procedures has also risen [66]. As pointed out above, genital wound repair differs in many ways from the skin of other body parts, implying that genital cutaneous surgery should only be performed by experts with profound knowledge of the anatomy and wound healing physiology of the genitalia.

Unlike non-genital wounds, acute genital wounds heal quickly. Massive wound infections, such as Fournier gangrene, are commonly only found in immune compromised patients or in association with diabetes. So what is magic about genital skin? Differences in three key features govern enhanced genital skin wound repair with reduced scarring, i.e., biomechanics, the inflammatory cellular response and the ability of intracrine sex hormone production.

Genital skin morphology resembles fur bearing animal skin with subcutaneous tissue devoid of fat but with a so-called fascia instead—reminiscent of the carnosus muscle. We showed an increased number of elastic fibers in the genital dermis compared to that of non-genital skin. Fast wound closure by wound margin contraction is found in many fur bearing animals, with highly elastic skin compared to human skin [67,68]. An interesting area for further studies would be wound closure kinetics of the outer genital organs. Excess of skin tissue, which is mobile due to flexible subcutaneous anchorage and that can stretch due to high tissue elasticity, may support fast defect coverage and wound closure as seen in animals. Increased tissue tension and the presence of transforming growth factor (TGF)-β trigger hypertrophic scarring in human extra-genital skin [21,69,70]. Presumably, unimpaired genital scarring is due to different biomechanical characteristics of the genital skin. For example, low tissue tension in genitalia originates from: i. the excess of skin tissue present in genitalia, a prerequisite for fast volume changes; ii. flexible anchorage of the cutaneous tissue to underlying structures (e.g., swelling bodies which are soft most of the time); iii. abundance of elastic fibers in dermis and multiple fasciae [11]. Lack of strong and permanent tissue tension caused by underlying structures such as bone or cartilage may be the reason for biomechanical differences that influence cellular behavior in genital tissue remodeling and scarring.

All surfaces of the outer genitalia are inhabited by a multitude of microbia normally not found in extra-genital skin. In order to deal with this colonialization, genital skin cells display a different immune response compared to other sites. Cellular expression of high levels of antimicrobial peptides (AMPs) and defensins protect against bacterial penetration into deeper and more vulnerable skin layers. Unfortunately, research and published literature on genital wound repair and the inflammatory reaction is scarce. However, there are similarities between genital skin and the oral cavity. The anatomical structures inside the vulva, e.g., small labia, clitoral hood, clitoris and vagina, as well as the corresponding male organs, e.g., inner sheath of the foreskin and glans, are non-keratinizing mucous epithelia, as the inner lining of the oral cavity is. Hence, comparison between both anatomical locations is legitimate. In mucosal wounds in mouse, pig and human, macrophage numbers were reduced [71,72] with less cytokine production in early wound healing followed by less scarring. Gene expression during wound repair is strongly different in mucosal wounds compared to skin with less expression of pro-inflammatory genetic elements in mucosa, compared to skin [73]. It is speculative whether genital mucous membranes contain a higher amount of stem cells—as the oral cavity does [74]—in comparison to normal skin. Oral stem cells dampen the inflammatory response to trauma [75] and are antibacterial [76]—features that may also apply to genital mucosal membranes. This highly interesting aspect has to be elucidated in future scientific studies. Effective skin wound healing is characterized by the conversion of M1 to M2 macrophages [77,78], which produce less pro-inflammatory cytokines. Cutaneous scarring is characterized by the effect of pro-fibrotic TGF-β and α-smooth-muscle actin [11] with myofibroblast formation [70]. Less inflammation and faster wound healing with almost no scarring as seen in oral and genital mucous membranes might be due to the influence of local stem cells which induce a faster macrophage polarization to M2 cells, which produce less inflammatory cytokines, followed by enhanced wound healing and reduced scarring [79].

The cellular inflammatory response to trauma or infection is of central importance for the wound healing process, including scarring. Skin keratinocytes secrete a variety of pro-inflammatory growth factors after wounding to attract immune cells for the initiation of the repair process [80,81,82,83]. This is in contrast with vaginal epithelial cells that secrete less IL-1β and no TNF-α after injury, with less scarring [84]. In a study comparing oral and epidermal keratinocytes, higher IL-6 and TNF-α secretion was found in challenged skin keratinocytes with prolonged wound closure in contrast to faster wound closure by oral keratinocytes [85]. Obviously, a quickly resolving inflammatory response by local cells governs repair processes, with a high resemblance between genital and oral epithelial cells.

Sex hormone responsiveness of genital tissues is crucial for an uncompromised development of the outer genitalia [86,87]. Androgen and estrogen receptors are found in pre- and postnatal genital skin and connective tissues [86,87,88,89]. A higher expression of AR and ER was noticed in small and big labia compared to pubic skin [88]. Skin is an important source for extragonadal sex hormone production. Genital fibroblasts produced significantly more DHT in comparison to non-genital skin cells [90,91], whereas DHT degradation was similar. Aromatase converts androgen precursors into estrogen [48], genital skin being important for estrogen production [92]. Stimulation of fibroblasts with cortisol increased genital aromatase activity, with higher estrogen production [46]. Obviously, hormonal sensitivity and intracrine hormone production of genital cells are present even before birth and have a pivotal role in maintaining the skin’s physiological homeostasis. Interestingly, no differences were found in genital AR or ER expression between pre- and postmenopausal women [88], showing that reduced gonadal estrogen production is mainly responsible for cutaneous changes during menopause.

Skin cells are sex hormone sensitive and convert precursors into their active counterparts. Skin wound repair and scarring are influenced by the hormonal status of the individual as shown above. In contrast to testosterone, estrogens facilitate wound closure [32]. Estrogens induce contraction in fibroblasts without myofibroblast induction [33]. Furthermore, estrogens reduce cellular oxidative stress, cell death and increase cell migration and collagen deposition [44]. Estrogen deficiency during menopause is characterized by skin atrophy, higher oxidative cellular stress and increased inflammatory reaction to exogenous stimuli [44]. Vaginal estrogen application can reverse the menopausal symptoms [93]. Furthermore, it is clinical practice to add estrogen creams to the vulva or vagina after wounding or genital surgery including gender reassignment surgery to enhance wound healing. Further studies on intracrine genital hormone production are needed to elucidate the importance of extra-gonadal sex hormone production for enhanced tissue repair and regeneration.

## 4. Conclusions

Skin wound repair of the outer genitalia differs from other areas of the body in that it proceeds faster, with less inflammation and almost no scarring. Lower tissue tension, higher content of elastic fibers, intracrine hormone production and faster resolution of the post-traumatic inflammatory cellular reaction in a moist healing environment are the main causes for superior genital wound healing in contrast to extra-genital skin.

## Figures and Tables

**Figure 1 medsci-10-00023-f001:**
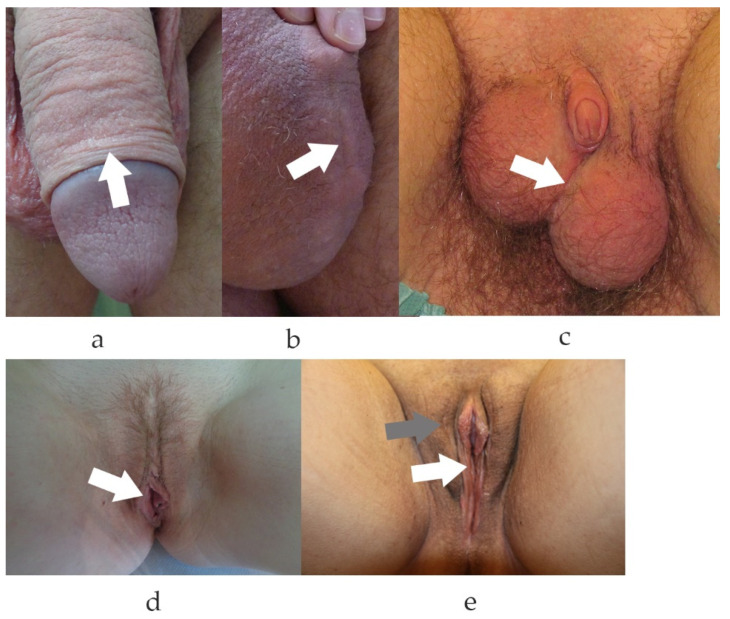
Genital scars of male, female and gender reassigned genitalia. (**a**). Typical scar after circumcision. Note, the scar is only visible due to different pigmentation of the inner and outer part of the penile foreskin. (**b**). Scrotal scar after abscess excision. Visible scar after prolonged inflammatory phase. (**c**). Scarring after female-to-male reassignment surgery with neo-scrotal sac formed from the major labia and metoidioplasty. (**d**). Scarring after surgical reduction plasty of the small labia. (**e**). Genital aspect and scarring after male-to-female reassignment surgery. In (**e**), grey arrow depicts the scar to the great labia (prior scrotal skin), the white arrow to the small labia (prior penile skin). (**a**–**e**). White arrows depict the scars.

**Figure 2 medsci-10-00023-f002:**
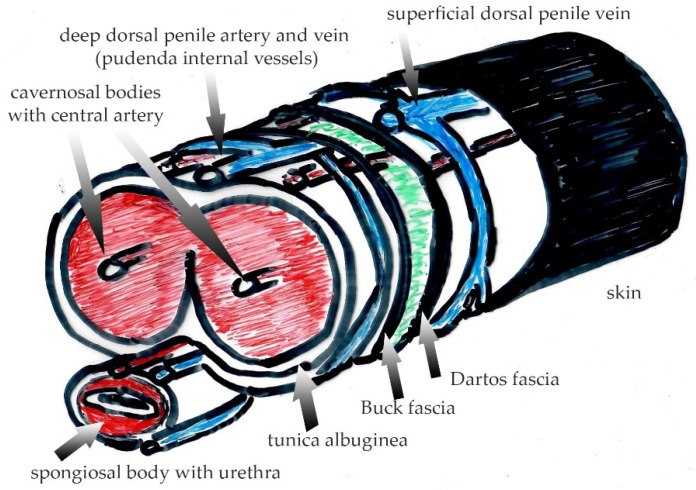
Penile anatomy with different layers separated by the Dartos’ and Buck’ fasciae. Note that innervation and blood supply to the outer skin is provided by the genitofemoral nerve and external pudendal vessels whereas the glans and the inner sheath of the foreskin are supplied by the dorsal penile nerve (originating from the pudendal nerve) and by the dorsal penile vessels (originating form the internal pudendal arteries and veins). Green color depicts the space between Dartos and Buck fascia with loose connective tissue intertwined between both fasciae, which enables unimpaired skin movement during erectile volume shifts.

**Figure 3 medsci-10-00023-f003:**
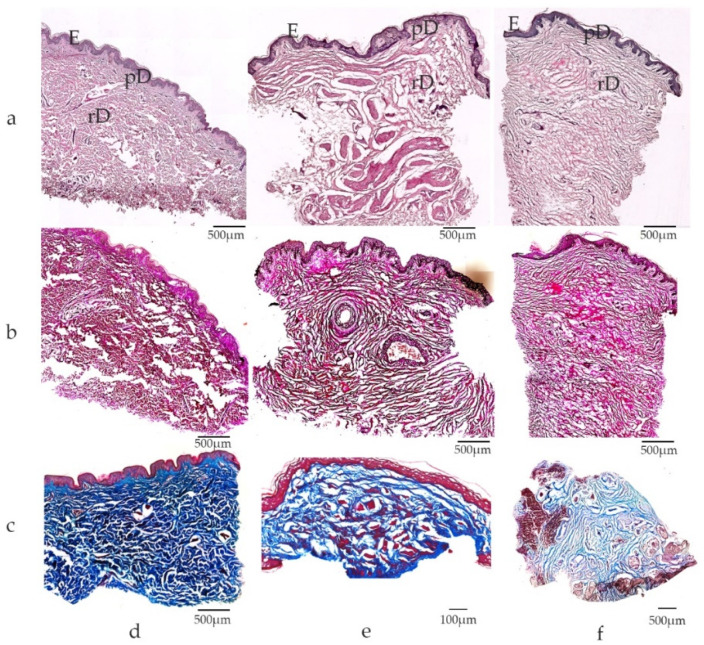
Microscopic view of extra-genital and genital skin specimens: (**a**) Hematoxylin-Eosin staining; (**b**) Elastica-van-Gieson staining; (**c**) Masson-Trichrome staining; skin samples from (**d**) extra-genital area ((**a**,**b**): arm, c breast); (**e**) penis; (**f**) small labia. Note differences in dermal composition between extra-genital and genital skin, with dense dermal tissue in skin derived from the upper arm in contrast to lose connective tissue in genital skin. Elastic fibers are visualized by Elastica-van-Gieson staining with collagen in red and elastic fibers in black. Elastic fibers are abundant in genital skin with long, perpendicular arrangement of fibers in genital dermal tissue. (**c**) Dense connective tissue (blue) is found in extra-genital skin (breast) in contrast to loosely packed connective tissue in the dermis of the penis (**e**) and small labia (**f**). E epidermis, pD papillary dermis, rD reticular dermis; scale bars depicted for each panel.

**Figure 4 medsci-10-00023-f004:**
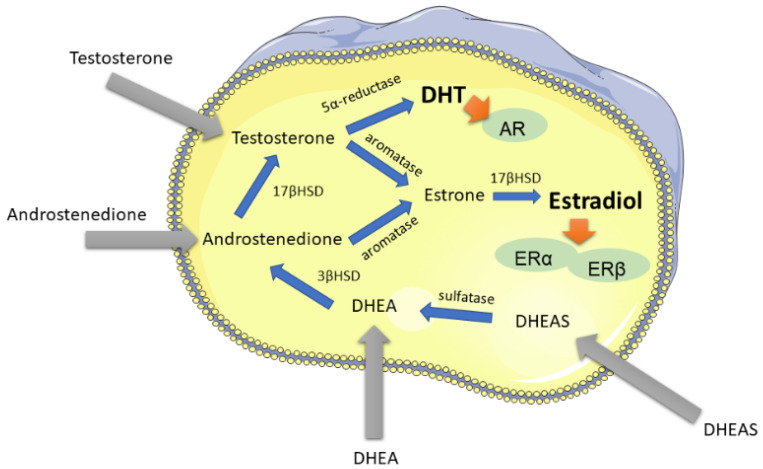
Schematic overview of intracrine hormonal processing pathways in non-gonadal peripheral cells. Steroid precursors such as DHEA, DHEAS, androstenedione or testosterone originate from the adrenal cortex. After cellular uptake, precursor forms are processed to more potent sex steroid hormones, e.g., DHT or estradiol for further androgen or estrogen receptor binding and intracrine nuclear signaling. DHEA dehydroepiandrosterone, DHT dihydrotestosterone, 17βHSD 17-beta-hydroxysteroid dehydrogenase, AR androgen receptor, ER estrogen receptor.

**Table 1 medsci-10-00023-t001:** Skin structure of homologous male and female outer genitalia (modified after Mirastschijski et al. [11]).

	Male	Female	Histological Features of the Skin
Penis/small labia	Penile shaft skin	Labia minores	Penis: keratinizing epithelium, presence of Dartos fascia (Fascia penis superficialis) Labia: outer aspect with thin keratinizing epithelium; inner aspect: mucous membrane Similar for penis and labia: absence of hair and fat; abundance of elastic fibers
Glans	penis	clitoridis	Multilayered, non-keratinizing epithelium
Foreskin	penis	clitoridis	Outside: cornified epithelium Inside: mucous membrane; no fat
Frenulum	Frenulum penis	Frenula clitoridis (paired)	Mucous epithelium, no subcutaneous fat
Scrotum/big labia	Scrotum	Labia majores	Hair bearing epidermis (labia: only outer surface), epidermal cornified layer Labia: subcutaneous fat layer and smooth muscle cells; reduction of fat amount with aging Scrotum: fat localized to scrotal wall and between scrotal septa in obese men, contractile Tunica Dartos with smooth muscle cells and myofibroblasts

## Data Availability

Not applicable.

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
