# Peer review of "Genital Wound Repair and Scarring"

_medsci, 2022, doi:10.3390/medsci10020023_

Round 1

Reviewer 1 Report

Genital wound repair is interesting issue. Many of the results, description and materials (IRB 2018-157) in this review paper are identical to the author’s 2020 publication in Plastic and Aesthetic Research, “Wound repair and scarring of genital skin”. Also, there are no citations in Figure 2. The “Biochemical skin composition, microstructure and biomechanics” in the Results is not sufficient to understand the biochemical composition of gential skin compared to extra-genital skin. References related to genital wounds such as Indian J Plast Surg. 2012 May-Aug; 45(2): 352–363, Female Pelvic Med Reconstr Surg. Nov/Dec 2018;24(6):419-423, Am J Clin Dermatol. 2018 Oct;19(5):695-706. doi: 10.1007/s40257-018-0364-7, etc. should also be included and cited more deeply.

Author Response

Medical Sciences

Manuscript ID: medsci-1289814

                                                                       Bremen, Oct. 31st, 2021

Dear ladies and gentlemen,

Please find below the reply to reviewer no 1. Comments and suggestions are appreciated. Changes were made accordingly which contribute to the amelioration of the manuscript. On behalf of the co-authors, I would like to thank reviewer 1 for the suggestions. Please find the reply and changes made to the manuscript highlightened in yellow below.

Original comments by reviewer no1:

Genital wound repair is interesting issue. Many of the results, description and materials (IRB 2018-157) in this review paper are identical to the author’s 2020 publication in Plastic and Aesthetic Research, “Wound repair and scarring of genital skin”. Also, there are no citations in Figure 2. The “Biochemical skin composition, microstructure and biomechanics” in the Results is not sufficient to understand the biochemical composition of gential skin compared to extra-genital skin. References related to genital wounds such as Indian J Plast Surg. 2012 May-Aug; 45(2): 352–363, Female Pelvic Med Reconstr Surg. Nov/Dec 2018;24(6):419-423, Am J Clin Dermatol. 2018 Oct;19(5):695-706. doi: 10.1007/s40257-018-0364-7, etc. should also be included and cited more deeply.

Reply to reviewer no1:

On behalf of me and my co-authors, I want to thank reviewer no1 for the comments. In comparison to the mentioned publications, the patient cohort has been extended specifically in the aesthetic cohort.

Reviewer 1: Also, there are no citations in Figure 2.

Answer: As stated by reviewer no1, there are indeed no citation in the legend of Figure 2. However, there are abundant citations in the text referring to Figure 2. I added the original text of the manuscript with citations highlightened in yellow. No changes were added to the text.

3.3. Macroscopic and microscopic anatomy of genital skin

Penile and labial (small labia and inner aspect of the big labia) skin is devoid of hair follicles, sebaceous and sweat glands in contrast to normal skin. The glans penis and clitoridis, the inner part of the foreskin (both sexes) as well as the small labia are covered by non-keratinizing epithelium, and thus resembles the oral cavity. There are additional many parallels between genital and oral wound repair as well and will be discussed later. The genital dermis is rich in loosely packed elastic fibers (Fig. 2) to enable cutaneous adaptation to volume changes during erection, genital intercourse or childbirth. Of note, the penile size increases during erection within seconds with an average of about 1.4 to 1.6 fold in length (unpublished data, [1]) and about 1.3 fold in circumference [1, 2]. Excess in skin with high elasticity and several fasciae allow for fast volume changes and tissue extension.

Reviewer 1: The “Biochemical skin composition, microstructure and biomechanics” in the Results is not sufficient to understand the biochemical composition of gential skin compared to extra-genital skin.

Answer: We thank the reviewer for this remark. Indeed, the paragraph consists predominantly of anatomical explanation for differences. Biochemical features were added to the text (pages 4 and 5) and highlightened in yellow.

The biochemical composition of the skin and its different layers is well established ({Uitto, 1989 #4955}{Pfisterer, 2021 #4956}). The dermal extracellular matrix consists predominantly of different types of collagens, followed by elastic fibers, glycoproteins and proteoglycans. Amongst collagens, type I collagen is abundantly produced by dermal fibroblasts in adult skin. Fibroblasts are the predominant cell type in the connective tissue that produce various extracellular matrix proteins, remodel the ECM constantly and are important for wound repair. Because tissues of different origin are exposed to tissue-specific biomechanical cues, fibroblasts originating from various body sites behave differently. Human dermal fibroblasts were very sensitive to stiff matrices whereas foreskin fibroblasts did not increase the production of collagen type III, matrix metalloproteinase (MMP)-2 or tissue inhibitor of MMP-2 ({Karamichos, 2007 #4957}) in response to mechanical strain. Of note, cellular MMP secretion is crucial for tissue remodeling and cell migration in skin wound repair ({Mirastschijski, 2004 #4322;Mirastschijski, 2002 #4327;Mirastschijski, 2010 #4311}).

Elastic fibers comprise elastin and microfibrils. While collagen fibers contribute to the tensile strength of the dermis, elastin is important for tissue elasticity and protects against tearing stresses {Pfisterer, 2021 #4956}. Interestingly, the percentage of elastic fibers varies between skin tissue of various sites. While dermal elastin content is around 2% to 8% ({Heinz, 2016 #4958}), genital skin contains up to 29% elastin ({Andrade, 2012 #4959}). Both elastic fibers and collagens are degraded by various MMPs {Giannandrea, 2014 #3227;Rohani, 2015 #3155}. The fragmentation of elastic fibers is found in aging {Mora Huertas, 2016 #4960} and fibrotic diseases such as lichen sclerosus {Romppanen, 1987 #4961}. According to the literature, we found in our studies abundant elastic fibers in genital tissue in contrast to non-genital skin (Fig. 2).

Reviewer no1: References related to genital wounds such as Indian J Plast Surg. 2012 May-Aug; 45(2): 352–363, Female Pelvic Med Reconstr Surg. Nov/Dec 2018;24(6):419-423, Am J Clin Dermatol. 2018 Oct;19(5):695-706. doi: 10.1007/s40257-018-0364-7, etc. should also be included and cited more deeply.

Answer: We thank the reviewer for the provision of the highly interesting references. The named references were discussed according to the subject and added to the text with highlightening in yellow.

  1. Indian J Plast Surg. 2012 May-Aug; 45(2): 352–363: manuscript page 2:

Fournier gangrene is a life-threatening disease. This disease occurs with an incidence of 1.6/100.000 patients and is mostly present in men (10:1) [6]. Thorough surgical debridement of necrotic tissue with simultaneous antibiotic treatment is the only sufficient cure. This is followed by a massive tissue loss and subsequent plastic-surgical reconstruction [7]. In general, plastic surgical treatment follows an algorithm for defect coverage of various sizes. For perineal wounds such an algorithm was published by Sharma et al 2012 [8].

  1. Female Pelvic Med Reconstr Surg. Nov/Dec 2018;24(6):419-423: this article states wound repair of the vaginal wall. Since the epithelium of the vagina differs enormously from the genital skin, vaginal repair was not an issue in our manuscript. Therefore, there is no passage in the manuscript dealing with vaginal wound repair. As a consequence, this reference could not be inserted due to lack of relevance for the current manuscript.

  1. Am J Clin Dermatol. 2018 Oct;19(5):695-706. doi: 10.1007/s40257-018-0364-7: page 3.

Chronic genital wounds are present in various dermatological diseases, e.g. lichen sclerosus et atrophicus (LSC). LSC is an autoimmune disease with chronic inflammation of mucous surfaces of the anogenital area [9]. LSC affects both sexes and can lead to severe scarring and tissue shrinkage when untreated [9].

Overall, we like to thank both reviewers for their substantial input that led to an amelioration of the present manuscript.

Kind regards,

Ursula Mirastschijski

for all co-authors

Reviewer 2 Report

The review “Genital wound repair and scarring” by Ursula Mirastschijski et al. provides and overview of genital skin wound healing and pathologies and aims to compare the morphology, physiology and wound repair of genital and extra-genital skin. Although surgery involving genital skin is a common practice throughout the world, very little literature is available on scar formation of the genital skin. With the interest in genital esthetic surgery on the rise, an overview of current knowledge on genital skin wound healing and scarring is of great interest to specialists in this field.

Broad comments:

Genital surgery performed in a sterile clinical setting can hardly be compared to male circumcision practice in rural communities with poor hygiene and no follow-up. Those completely different settings should be clearly identified as such (as well as indicate whether certain conclusions are based on and/or apply to sterile or non-sterile genital skin wound healing).

The introduction mentions several genital diseases associated with inflammation. How does this impact normal genital wound healing and/or scarring?

Section 2 (especially when the fascia layers comes into play) is a little hard to comprehend without an illustration. Adding corresponding annotations to the histological images of figure 2 and/or adding the different fascia layers in table 1 would also help.

In section 3.3: how does or would excess skin with high elasticity and several fasciae (mechanistically) impact genital wound healing?

Section 3.4 check references are correct, both [17] and [18] only report on non-genital skin. Considering the aim of the paper, clearly indicate differences between genital and extra-genital skin. If no data is available in literature, clearly identify speculations and hypotheses as such.

Section 3.5 (first 2 paragraphs) are redundant. Please elaborate/include a (mechanistic) explanation on how hormone responsiveness and lack of mechanical tension affects genital wound healing/scarring.

Section 4 is redundant, please revise and avoid repetition.

Overall: many references seem incorrect – please check and add appropriate references.

Specific comments:

Figure 1: The scars are difficult to appreciate. please add arrow(head)s indicating the scar(s). Picture a is already published elsewhere [8].

Figure 2:

  • add scale bar in all panels.
  • c,e is diffenrent magnification than c,f and c,g. please use same magnification in comparative pictures.
  • increase contrast of the black arrow (e.g. change color) in panel c as it is hardly visible in black.
  • Panels c(e,f,g) and d(e+f) in figure 2 are exact duplicates of ref [8].

Figure 3: An illustration of the intracrine estrogen signaling would be more appropriate/usefull in supporting the hypothesis/conclusion that intracrine hormone production causes superior genital skin wound healing.

Line 115-117 (absence of bona fide fascia etc) seems overstated and requires proper reference.

Author Response

Medical Sciences

Manuscript ID: medsci-1289814

                                                                       Bremen, Oct. 31st, 2021

Dear ladies and gentlemen,

Please find below the reply to reviewer no 2. Comments and suggestions are appreciated. Changes were made accordingly which contribute to the amelioration of the manuscript. On behalf of the co-authors, I would like to thank reviewer 1 for the suggestions. Please find the reply and changes made to the manuscript highlightened in yellow below.

Original comments by reviewer no2:

Broad comments:

Genital surgery performed in a sterile clinical setting can hardly be compared to male circumcision practice in rural communities with poor hygiene and no follow-up. Those completely different settings should be clearly identified as such (as well as indicate whether certain conclusions are based on and/or apply to sterile or non-sterile genital skin wound healing).

The introduction mentions several genital diseases associated with inflammation. How does this impact normal genital wound healing and/or scarring?

Answer: we thank reviewer 2 for the comment and added the following text to the introduction with highlightening.

Chronic genital wounds are present in various dermatological diseases, e.g. lichen sclerosus et atrophicus (LSC). LSC is an autoimmune disease with chronic inflammation of mucous surfaces of the anogenital area [9]. LSC affects both sexes and can lead to severe scarring and tissue shrinkage when untreated [9].

The genital area hosts urogenital and anal orifices are colonized by resident microbia in a moist environment. Skin tissue is found in abundance with highly elastic skin that is loosely attached to the pelvic bone. Hence, genital skin differs in several aspects from non-genital skin. The inflammatory reaction and hormonal responsiveness of genital skin differ in comparison to other tissues by cellular responses being adapted to the local microflora with fast resolution of inflammatory responses and wound closure.

Section 2 (especially when the fascia layers comes into play) is a little hard to comprehend without an illustration. Adding corresponding annotations to the histological images of figure 2 and/or adding the different fascia layers in table 1 would also help.

Answer: The penile anatomy is quite complex, indeed. Therefore, we added another figure (now Fig. 2) on page 5 showing the different penile fascia and structures in detail. The figure legend reads as follows:

Figure 2: Penile anatomy with different layers separated by the Dartos’ and Buck’ fasciae. Of note, innerveration and blood supply of the outer skin is provided by the genitofemoral nerve and external pudendal vessels whereas the glans and the inner sheath of the foreskin are supplied by the dorsal penile nerve (originating from the pudendal nerve) and by the dorsal penile vessels (originating form the internal pudendal arteries and veins). Green color depicts the space between Dartos and Buck fascia with loose connective tissue intertwining between both fasciae that enables unimpaired skin movement during erectile volume shifts.

In section 3.3: how does or would excess skin with high elasticity and several fasciae (mechanistically) impact genital wound healing?

Answer: The comment of reviewer 2 is very important. Therefore, more information was added on page 7.

Excess of highly elastic and flexible skin is an important feature of the outer genitalia in both sexes. This phenomenon is highly important not only for fast volume changes during sexual intercourse, thermal regulation of the testes or childbirth but also for reconstructive purposes. In case of traumatic tissue loss, half of the scortal sac can be reconstructed by the remaining scrotal tissue [10]. Furthermore, excess skin is useful for defect coverage of the penile shaft in obese men [35, 36], after trauma [37] or tumor excision [28, 37] or for genital reassignment surgery [38].

Section 3.4 check references are correct, both [17] and [18] only report on non-genital skin. Considering the aim of the paper, clearly indicate differences between genital and extra-genital skin. If no data is available in literature, clearly identify speculations and hypotheses as such.

Answer: This comment required substantial revision of the manuscript. Section 3.4 and references were revised and changed as highlightened.

3.4. Genital autocrine hormone production and endocrine responsivness

Skin wound repair is influenced by hormones. While estrogens accelerate wound healing [29], testosterone has the opposite effect [30]. In postmenopausal women and in men, the adipose tissue and the skin are the major source of peripheral estrogen production. Skin cells can produce their own sex hormones by converting renal precursor peptides such as dehydroepiandrosterone (DHEA) into estrone by aromatase or into testosterone by 17β-hydroxysteroid-dehydrogenase (17β-HSD) (Fig. 4). Estrogens enhance epidermal keratinocyte and dermal fibroblast migration [31]. In non-genital skin, steroid hormone addition stimulated fibroblast contraction without alpha-smooth-muscle expression [31] whereas the contractility of tunica albuginea fibroblasts was directly enhanced by estradiol with concomitant reduction of myofibroblasts [32]. Proliferation was increased in non-genital [33] and genital [34] keratinocytes by estradiol via estrogen receptor (ER) signaling. ERα and ERβ receptors are expressed both in non-genital and genital skin [34-36]. The anti-inflammatory effect of estrogens has been extensively demonstrated for non-genital skin [30, 37]. In genital keratinocytes, anti-inflammatory estradiol reduced TNFα induced RANTES secretion, the effects of monocyte chemoattractant protein-1 and inhibited the effect of IFNγ [38-40].

Intracrine production of estrogens requires the enzyme aromatase that converts C19 steroid hormone precursors, e.g. androstenedione, or testosterone into estrone. In contrast to dermal fibroblasts, keratinocytes increase their aromatase secretion 400fold upon mechanical wounding with subsequently enhanced estrogen levels [31]. Estrogens reduce the inflammatory reaction [41], the oxidative stress and cellular apoptosis during wound repair [10, 42]. Aromatase activity is age-dependent [43] and expressed in genital and non-genital fibroblasts [44, 45]. In postmenopausal women, aromatase is upregulated in skin fibroblasts [46] and catalyzes estrogen production via testosterone aromatization [45]. Alternatively, testosterone is irreversibly converted into dihydrotestosterone (DHT) by 5α-reductase [47]. Interestingly, testosterone metabolism occurs much faster in genital skin fibroblasts compared to non-genital cells [48]. 5α-reductase is expressed in genital and non-genital skin [49] with age-dependent decrease of the type 2 but not type 1 enzyme in human foreskin [50]. DHT acts via binding to the intracellular androgen receptor (AR) which is found in both, genital and non-genital skin [45]. DHT exposure increased binding to AR in genital skin fibroblasts in a dose- and time-dependent manner [51]. Significantly higher levels of AR, 5α-reductase and 17β-HSD were found in scrotal fibroblasts compared to blood cells [49]. Further information on hormone production and sex hormone receptor expression between non-genital and genital skin are found in detail elsewhere [10, 52].

Section 3.5 (first 2 paragraphs) are redundant. Please elaborate/include a (mechanistic) explanation on how hormone responsiveness and lack of mechanical tension affects genital wound healing/scarring.

Answer: The second paragraph was omitted. A mechanistic explanation was included.

Excess of highly elastic and flexible skin is an important feature of the outer genitalia in both sexes. This phenomenon is highly important not only for fast volume changes during sexual intercourse, thermal regulation of the testes or childbirth but also for reconstructive purposes. In case of traumatic tissue loss, half of the scortal sac can be reconstructed by the remaining scrotal tissue [10]. Furthermore, excess skin is useful for defect coverage of the penile shaft in obese men [35, 36], after trauma [37] or tumor excision [28, 37] or for genital reassignment surgery [38].

Section 4 is redundant, please revise and avoid repetition.

Overall: many references seem incorrect – please check and add appropriate references.

Specific comments:

Figure 1: The scars are difficult to appreciate. please add arrow(head)s indicating the scar(s). Picture a is already published elsewhere [8].

Answer: Fig. 1a was exchanged, arrows were added. The legend reads now as follows:

Figure 1. Genital scars of male, female and gender reassigned genitalia. a. Typical scar after circumcision. Note, the scar is solely visible due to different pigmentation of the inner and outer part of the penile foreskin. b. Scrotal scar after abscess excision. Visible scar after prolonged inflammatory phase. c. Scarring after female-to-male reassignment surgery with neo-scrotal sac formed from the major labia and metoidioplasty. d. Scarring after surgical reduction plasty of the small labia. e. Genital aspect and scarring after male-to-female reassignment sugery. Arrows depicting the scars. e. grey arrow depicting scar to the great labia (prior scrotal skin), white arrow to the small labia (prior penile skin).

Figure 2:

  • add scale bar in all panels.
  • c,e is diffenrent magnification than c,f and c,g. please use same magnification in comparative pictures.
  • increase contrast of the black arrow (e.g. change color) in panel c as it is hardly visible in black.
  • Panels c(e,f,g) and d(e+f) in figure 2 are exact duplicates of ref [8].

Answer: Figs 3 c-e, c-f and c-g were omitted. Scale bars were added to each panel. The legend reads now as follows:

Figure 3. Microscopical view of extra-genital and genital skin specimen: (a) Hematoxylin-Eosin staining; (b) Elastica-van-Gieson staining; (c) Masson-Trichrome staining; Skin samples from (d) extra-genital area (a-b: arm, c breast); (e) penis; (f) small labia. Note differences in dermal composition between extra-genital and genital skin with dense dermal tissue in skin derived from the upper arm in contrast to lose connective tissue in genital skin. Elastic fibers are visualized by Elastica-van-Gieson staining with collagen in red and elastic fibers in black. Abundant of elastic fibers are found in genital skin with long, perpendicular arrangement of fibers in genital dermal tissue. (c) Dense connective tissue (blue) is found in extra-genital skin (breast) in contrast to loosely packed connective tissue in the dermis of the penis (e) and small labia (f). E epidermis, pD papillary dermis, rD reticular Dermis; scale bars depicted for each panel.

Figure 3: An illustration of the intracrine estrogen signaling would be more appropriate/usefull in supporting the hypothesis/conclusion that intracrine hormone production causes superior genital skin wound healing.

Answer: As requested, the previous Fig. 3 was exchanged with a new Fig. 3 depicting intracrine signalling. The new legend reads as follows:

Figure 4. Schematic overview of intracrine hormonal processing pathways in non-gonadal peripheral cells. Steroid precursors such as DHEA, DHEAS, androstenedione or testosterone derive from the adrenal cortex. After cellular uptake, precursor forms are processed to more potent sex steroid hormones, e.g. DHT or estradiol for further androgen or estrogen receptor binding and intracrine nuclear signaling. DHEA dehydroepiandrosterone, DHT dihydrotestosterone, 17βHSD 17-beta-hydroxysteroid dehydrogenase, AR androgen receptor, ER estrogen receptor

Line 115-117 (absence of bona fide fascia etc) seems overstated and requires proper reference.

Answer: the term was omitted from the sentence.

Overall, we like to thank both reviewers for their substantial input that led to an amelioration of the present manuscript.

Kind regards,

Ursula Mirastschijski

for all co-authors

Reviewer 3 Report

Manuscript Number: medsci-1289814

Title: Genital wound repair and scarring

This review article by Dr. Rinkevich, a world-leading expert in tissue repair, and his colleagues is well written and summarizes nicely the differences between extra-genital and genital skin. The authors derive explanations for different kinetics and qualities of wound healing in extra-genital and genital skin. This article is of medical and scientific interest to dermatologists and researchers in the field of tissue repair. Minor points as listed below should be addressed by the reviewers.

Minor points:

  1. One paragraph is duplicated: Page 6, lines 149-153, and page 8, lines 207-211.
  1. Please check the line spacing. This differs from section to section.

  1. Page 2, line 51: Please insert “as” after “such”.

  1. Page 3, line 69: “The genital area hosts urogenital and anal orifices are colonized by resident microbia in a moist environment.” Please insert “that” after “orifices”.

  1. Page 4, line 105: Please replace “cornosus” with “carnosus”.

  1. Page 8, line 216: Please replace “macrophates” with “macrophages”.

Author Response

 Answer: We thank the reviewer for these observant remarks and are sorry for the spelling errors. All errors were corrected.

Reviewer 4 Report

The topic is very interesting but there are some points that should be improved. You should diveded the ulceration by etiology (inflammatory such as pyoderma gangrenosum, infectious , traumatic ...). The should develop better the topics of the phases of wound healing, the moist environment in wound healing, the role of tgf B, the treatments based on the etiology and the type of ulcers( TIME, PG- TIME, HS- TIME and more),the role of the biopsy and wound assessment (ultrahigh frequency ultrasound in dermatology and wounds)

Author Response

Answer: We thank the reviewer for the suggested changes. Pyoderma gangraenosum is found in extra-genital skin and TIME criteria were published for chronic extra-genital wound assessment and treatment. Since this article refers mainly to genital acute and chronic wound repair, addition of these features would expand beyond the subject. The topics on moist wound healing, TGF-beta function in scarring were added to the text and marked in green.

Round 2

Reviewer 1 Report

This paper contains very diverse results and massive efforts. However, it is very regrettable that there are several serious problems. It seems that the authors need a lot of time and effort to improve this article, and it is doubtful whether it will be possible. However, if it is improved, it will be an excellent article.

Author Response

Answer: All ticked items were improved as stated in colour in the text. New references were added, present references were revised. Aim of the project was added at the end of the introduction and marked in green. English was edited by two native speakers and changes marked in blue. We hope that the current version is suitable for publication.

Reviewer 2 Report

Although the authors made substantional revisions to the manuscript, I still have some major concerns, which are stated below.

Introduction: The introduction does not state the aim of the review. Please include this for readability.

The authors conclude: "Skin wound repair to the outer genitalia differs from other areas of the body in that it proceeds faster with less inflammation and almost no scarring."
Please provide (references to) clinical or experimental data that shows less inflammation in genital skin wound healing as compared to non-genital skin.

In line 215-216 (page 8) the authors state that "the inflammatory reaction after trauma is fast with rapid resolution due to conversion of M1 to M2 macrophates and reduced production of pro-inflammatory cytokines". It is not clear if this statement applies to wound healing in general, or genital skin wound healing specifically.  IF the latter, please provide proper references. 

The authors conlcude: "Less tissue tension, higher content of elastic fibers, intracrine hormone production and faster resolution of the post-traumatic inflammatory cellular reaction in a moist healing environment are the main causes for superior genital wound healing in contrast to extra-genital skin."
This conclusion is not (sufficiently) supported by the data provided. Please provide proper data and/or references. E.g. how is tissue tension different between genital and non genital skin and how does this impact (genital) skin wound healing? How is intracrine hormone signalling different in genital and non-genital skin (if at all)? How is the (cellular) inflammatory response different in genital and non-genital skin after wounding/during wound healing? 

Author Response

Introduction: The introduction does not state the aim of the review. Please include this for readability.

Answer: Aim of the project was added to the end of the introduction and marked in green.

The authors conclude: "Skin wound repair to the outer genitalia differs from other areas of the body in that it proceeds faster with less inflammation and almost no scarring."
Please provide (references to) clinical or experimental data that shows less inflammation in genital skin wound healing as compared to non-genital skin.

Answer:  Indeed, literature on inflammation in genital wounds is scarce. However, genital cutaneous tissue – especially small labia, clitoridal hood and the vagina, are epithelia with mucous membranes which strongly resemble the inner lining of the oral cavity. There is abundant literature on healing processes and inflammation of the oral skin. Changes were added to the text marked in green.

In line 215-216 (page 8) the authors state that "the inflammatory reaction after trauma is fast with rapid resolution due to conversion of M1 to M2 macrophates and reduced production of pro-inflammatory cytokines". It is not clear if this statement applies to wound healing in general, or genital skin wound healing specifically.  IF the latter, please provide proper references

 Answer: References were revised and new references added. Changes were marked in the text in green.

The authors conlcude: "Less tissue tension, higher content of elastic fibers, intracrine hormone production and faster resolution of the post-traumatic inflammatory cellular reaction in a moist healing environment are the main causes for superior genital wound healing in contrast to extra-genital skin."
This conclusion is not (sufficiently) supported by the data provided. Please provide proper data and/or references. E.g. how is tissue tension different between genital and non genital skin and how does this impact (genital) skin wound healing? How is intracrine hormone signalling different in genital and non-genital skin (if at all)? How is the (cellular) inflammatory response different in genital and non-genital skin after wounding/during wound healing? 

Answer: the complete Result and Discussion section have been revised and data / references added to support the current statement. Changes were marked in green.